# RNA fusion in human retinal development

**Wen Wang†, Xiao Zhang†, Ning Zhao, Ze-Hua Xu, Kangxin Jin\*, Zi-Bing Jin\***

Beijing Institute of Ophthalmology, Beijing Tongren Eye Center, Beijing Tongren Hospital, Capital Medical University, Beijing, China

**Abstract** Chimeric RNAs have been found in both cancerous and healthy human cells. They have regulatory effects on human stem/progenitor cell differentiation, stemness maintenance, and central nervous system development. However, whether they are present in human retinal cells and their physiological functions in the retinal development remain unknown. Based on the human embryonic stem cell-derived retinal organoids (ROs) spanning from days 0 to 120, we present the expression atlas of chimeric RNAs throughout the developing ROs. We confirmed the existence of some common chimeric RNAs and also discovered many novel chimeric RNAs during retinal development. We focused on *CTNNBIP1-CLSTN1* (*CTCL*) whose downregulation caused precocious neuronal differentiation and a marked reduction of neural progenitors in human cerebral organoids. *CTCL* is universally present in human retinas, ROs, and retinal cell lines, and its loss-of-function biases the progenitor cells toward retinal pigment epithelial cell fate at the expense of retinal cells. Together, this work provides a landscape of chimeric RNAs and reveals evidence for their critical role in human retinal development.

## Editor's evaluation

The primary goal of this paper is to profile expression of chimeric RNAs in human retinal organoids throughout development. The fundamental findings provide an important step forward in understanding retinogenesis specifically and neurogenesis, generally. The compelling evidence unveils a previously unrecognized significant role of chimeric RNA CTCL in human retinal development.

**\*For correspondence:**
jinkx@mail.ccmu.edu.cn (KJ);
jinzb502@ccmu.edu.cn (Z-BJ)

†These authors contributed
equally to this work

**Competing interest:** The authors declare that no competing interests exist.

## Introduction

The human retina is a laminar structure with a large number of different component cells that form morphologically and functionally distinct circuits. They work in parallel and in combination to produce complex visual output (*Hoon et al., 2014*). During retinogenesis, different subtypes of neurons are generated from the same group of retinal progenitor cells (RPCs) and precisely self-assemble into a functionally mature retina (*Masland, 2012*).

Dissecting the molecular mechanisms of human retinogenesis and functional maintenance has always been a focus and a challenging issue, which is particularly important for the treatment of human retinal diseases such as macular degeneration. A few groups have systematically elucidated the transcriptomics, chromatin accessibility, and proteomic dynamics during human and mouse retinogenesis (*Huang et al., 2023*), and comprehensively described the similarities and differences during this process. Lu et al. identified an unexpected role for *ATOH7* expression in the regulation of photoreceptor specification during late retinogenesis as well as the enriched bivalent modification of H3K4me3 and H3K27me3 in human but not in murine retinogenesis (*Lu et al., 2020*), further underscoring the limitations of using mouse models to study the human retina and transcriptional regulation in human retinogenesis.

Difficulties in obtaining normal human retinal tissue have hampered studies of human retinal development. The emergence of retinal organoids (ROs) could circumvent this problem (*Cheng et al., 2023*; *Jin et al., 2019*). Many groups have used ROs to conduct studies related to human retinal

development and have solved a basket of previously unsolvable problems, confirming the high similarity between ROs and human retina in terms of molecular features, cell types, and developmental rates (*Cowan et al., 2020*; *Lu et al., 2020*; *Xie et al., 2020*). Our group has extensive experience in RO culture and has conducted a series of studies using ROs (*Deng et al., 2018*; *Ma et al., 2022*; *Pan et al., 2020*; *Zhang et al., 2021*). For example, together with other groups, we have used ROs to confirm that retinoblastoma tumor cells are derived from cone precursor cells (*Clevers, 2016*; *Jin et al., 2019*; *Li et al., 2022*; *Liu et al., 2021*; *Liu et al., 2020*).

Chimeric RNAs are the ligation products of two or even more DNA or RNA sequences before or after transcription. They can be further translated into proteins in addition to their RNA form, increasing the richness of the transcriptome and proteome. Previous studies of chimeric RNAs have focused on tumors and larger tissues and organs, and found that chimeric RNAs are widely present in human normal and tumor tissues as part of the transcriptome and can regulate the activities of individual cells (*Elfman and Li, 2018*; *Hu et al., 2018*; *Mertens et al., 2015*; *Singh et al., 2020*). It has been shown that chimeric RNAs in normal human cortex increase with age and show differences in individual cells and tissues (*Mehani et al., 2020*), highlighting the critical role of chimeric RNAs in central nervous system (CNS) development and cell lineage maintenance. In 2021, Luo's group found that downregulation of *CTNNBIP1-CLSTN1* (*CTCL*) affects the growth of cerebral organoids, causing premature neuronal differentiation, and a marked reduction of neural progenitors (*Ou et al., 2021*). As retina is a part of the CNS, we hypothesize that *CTCL* is also present in the retina and plays an important role in human retinal development.

This study aims to investigate the molecular mechanism of human retinal development and function maintenance from a new perspective of chimeric RNAs. We used human embryonic stem cell

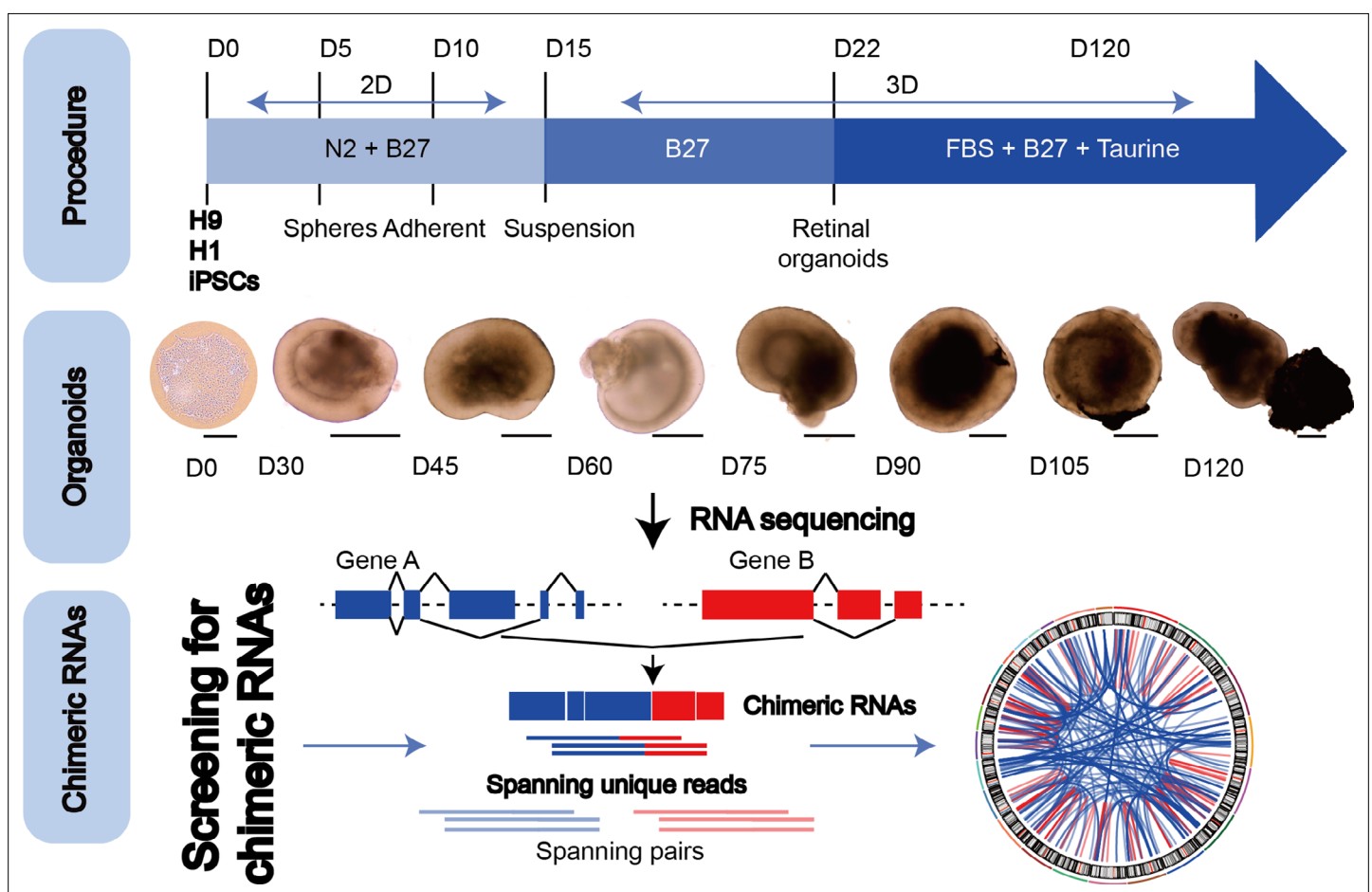

**Figure 1.** The general scheme of detection of chimeric RNAs. (Top panel) Procedure for the generation of human retinal organoids (ROs) from pluripotent stem cells. (Middle panel) D0–120 human ROs were used for bulk RNA sequencing. Scale bars: 200 µm (D0) and 400 µm (D30–D120). (Bottom panel) Illustration of chimeric RNA screening process and criteria by FusionCatcher.

(hESC)-derived ROs and performed RNA sequencing. We found that chimeric RNAs were expressed at all stages of RO development. Chimeric RNA *CTCL* is widely expressed in the human retina but not in the murine retina, and is critical for the differentiation of RPCs into neural retina (NR) or retinal pigment epithelium (RPE). RPCs with a relatively high level of *CTCL* expression will differentiate into NR, while those with a relatively low level of *CTCL* will differentiate toward RPE. These results provide important evidence for the physiological function of chimeric RNAs in human retinogenesis.

## Results

### Chimeric RNAs are present throughout the retinogenesis

To explore the chimeric RNAs in the developing human retina, we cultured ROs according to the methods described previously (*Kim et al., 2019*; *Lowe et al., 2016*). On day 0 (D0) of differentiation, hESCs in a good growth state were dissociated into individual cells and seeded at a certain cell density in V-shaped low-adhesion culture dishes so that the cells would aggregate into clusters to induce the neuroepithelial formation. At D30 of differentiation, the RO outermost layer with translucent NR structures was visible. As the differentiation time prolonged, the NR gradually became denser. At D105 of differentiation, ROs could be seen with the formation of RPE. At D120 of differentiation, the outer segments of photoreceptor cells were seen on the surface of ROs. We collected organoids from 0 to 120 days for RNA sequencing (*Figure 1*, top and middle panels). Using FusionCatcher software, we examined chimeric RNAs in human ROs covering eight developmental stages from D0, 30, 45, 60, 75, 90, and 105–120, corresponding to in vivo developmental stages D0, 30, 45, 60, 75, 90, 105, and 120 (*Cowan et al., 2020*), respectively. Each stage included two to three biological replicates (a total of *N* = 22), and each replicate contained approximately 100 ROs. To reduce false positives, splicing events were considered as positive chimeric RNA candidates only if spanning unique reads were detected at least once (*Figure 1*, bottom panel).

Chimeric RNAs were continuously expressed during RO development (*Figure 2A*). Chimeric events were categorized according to the locations of the parental genes on the chromosome, either inter- or intra-chromosomal. While the number of intra-chromosomal chimeric RNAs increased with the development of ROs ($r^2$ = 0.93, p = 0.00076, Pearson's correlation analysis), there was no such trend for inter-chromosomal chimeric RNAs (*Figure 2B,C*). Chimeric events can also be categorized according to the predicted effect (*Figure 3—figure supplement 1*). In the top three categories, the number of 'In-frame' chimeric RNAs was 11.2%, followed by 'CDS(truncated)_UTR' (10.7%) and 'UTR_UTR' (10.7%) (*Figure 3A*). We further classified chimeric events according to the type of their parental genes. The majority of chimeric RNAs were formed between two protein-coding genes (61.9%) (*Figure 3B*).

Next, we examined the motifs covering 10 bp sequences immediately upstream or downstream of the fusion site of parental genes and found that the canonical AG/GT donor–acceptor motif had the highest position weight (*Figure 3C*). In addition, we also found that the expression of the chimeric RNAs did not correlate with the expression of the parental genes (*Figure 3D*). These results were consistent with previous reports (*Ou et al., 2021*; *Singh et al., 2020*), indicating that the procedure for screening chimeric RNAs was reliable and that these chimeric RNAs do not arise randomly, suggesting that they may play a role in the normal development and physiological activity of the retina.

### *CTCL* is present in the human retina and at all the stages of RO development

*CTCL* has been shown to have a regulatory role in human cerebral development. Similarly, four isoforms of *CTCL*, joined by alternative splicing of the parent pre-mRNAs, were detected in the ROs by RNA-seq. In-frame *CTCL* can be further translated into chimeric proteins, and we think that it plays a more critical role than the other three isoforms. It was dynamically expressed during RO development and there were two obvious higher expression timepoints at D60 and D120 (*Figure 4A, B*). Retinal cells like cones, rods, and horizonal cells had been generated around D120, so we chose D60 when most retinal cells were progenitors. To further validate the presence of *CTCL* in the developing human ROs, we extracted total RNAs from D60 human ROs, followed by reverse transcription and Sanger sequencing using specific primers to amplify the fragments containing fusion sites between *CTNNBIP1* and *CLSTN1* (*Figure 4—source data 1*; *Figure 4—source data 2*; *Figure 4C,D* and Key

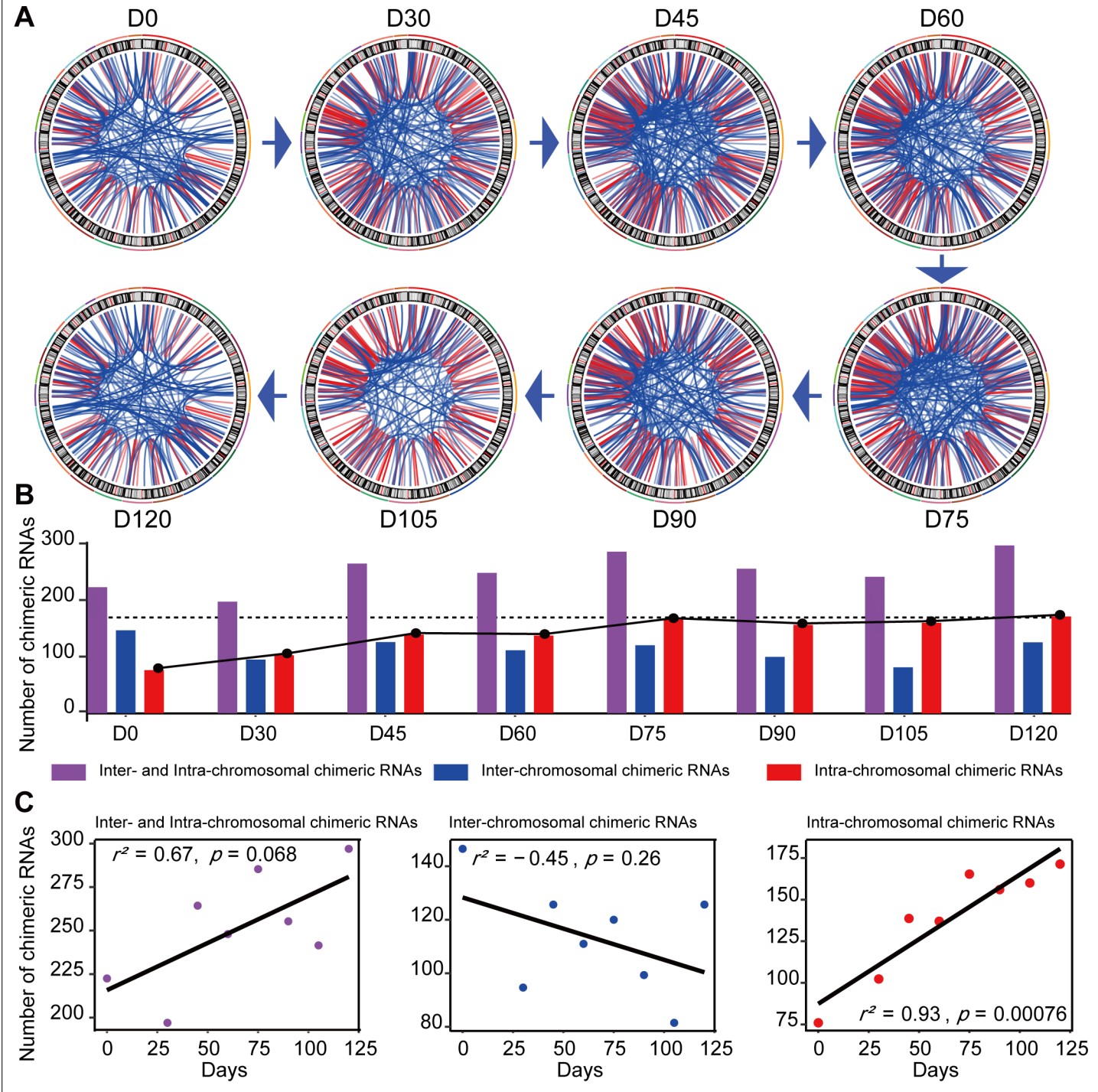

**Figure 2.** Expression of chimeric RNAs in the developing human retinal organoids. (**A**) Circos plots of genomic distribution of chimeric RNA parental genes observed in this work. Red lines indicate parental genes located in the same chromosome. Blue lines indicate parental genes located in different chromosomes. The outermost colored lines of circos plots represent chromosomes. (**B**) Types of chimeric RNAs based on parental genes' genomic distribution. (**C**) Pearson's correlation analysis of the number of chimeric RNAs and developmental stages.

resources table). At D60, the In-frame, UTR_CDS(truncated)-1, and CDS(truncated)Intronic *CTCL* isoforms were confirmed but UTR_CDS(truncated)-2 *CTCL* was not detected.

To further validate the widespread presence of the In-frame *CTCL* in vivo and in vitro, we examined it in various samples including mouse retinas, three human induced pluripotent stem cell (iPSC)-cultured ROs, and human retinas. Indeed, the In-frame *CTCL* was consistently identified in human

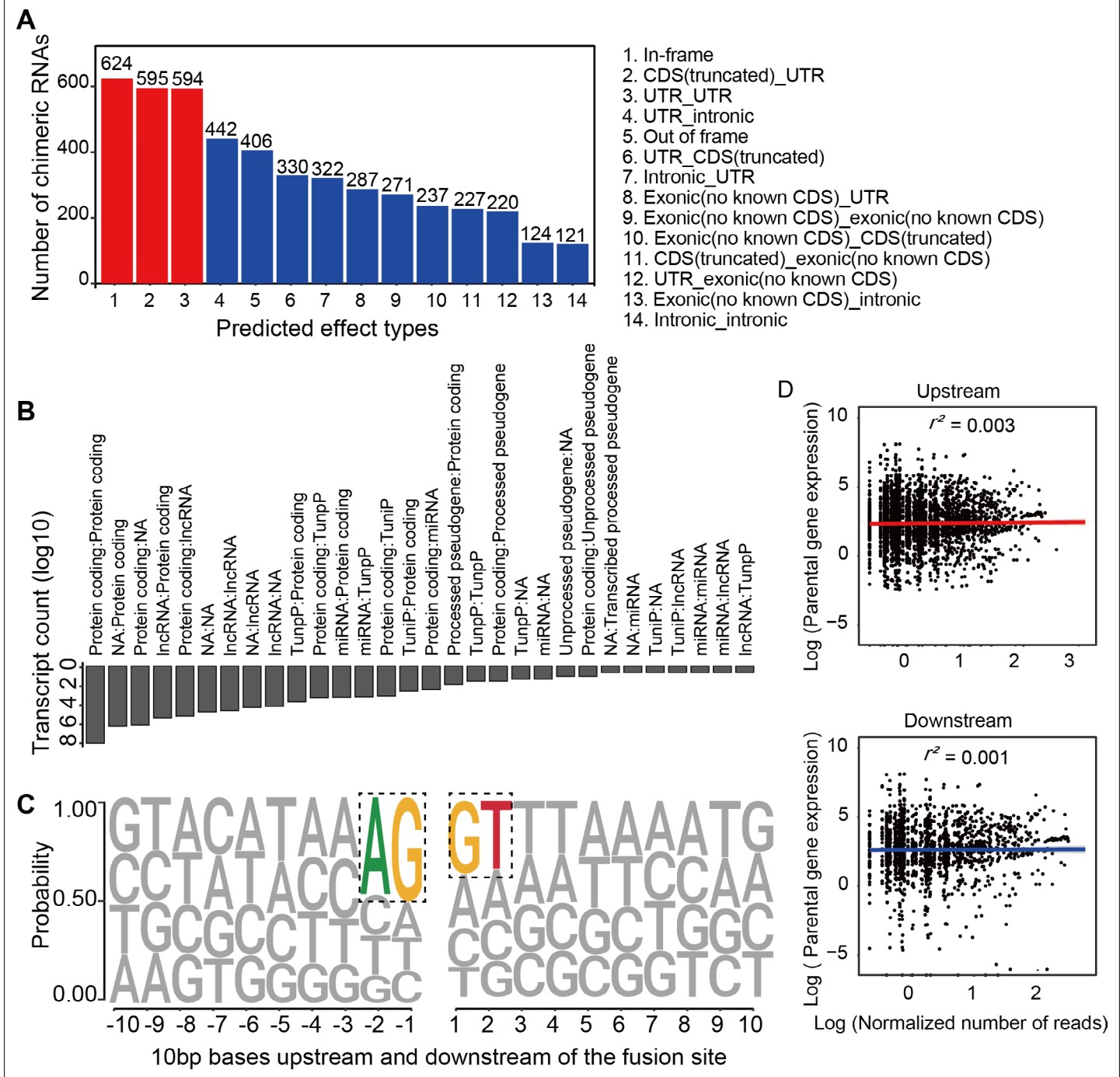

**Figure 3.** Characterization of chimeric RNAs in the developing human retinal organoids. (**A**) Top 14 types and corresponding numbers of chimeric RNAs based on predicted effects. (**B**) Biotype quantification of parental gene combinations in all samples. TunpP: transcribed unprocessed pseudogene; TuniP: transcribed unitary pseudogene. (**C**) Motifs consisting of 20 bp DNA sequences around the fusion site. (**D**) Spearman correlation analysis of expression level of chimeric RNAs and their parental genes, p > 0.05.

The online version of this article includes the following figure supplement(s) for figure 3:

**Figure supplement 1.** Types of chimeric RNAs based on predicted effect.

retinas, iPSCs1/iPSCs2/iPSCs3-derived ROs, and H9 cells, but not in the mouse retinas (*Figure 5—source data 1*; *Figure 5—source data 2*; *Figure 5—source data 3*; *Figure 5—source data 4*; *Figure 5*). Moreover, we also screened *CTCL* using the retinal transcriptome sequencing data available in the Gene Expression Omnibus (GEO) database, all of the nine human retinal samples yielded

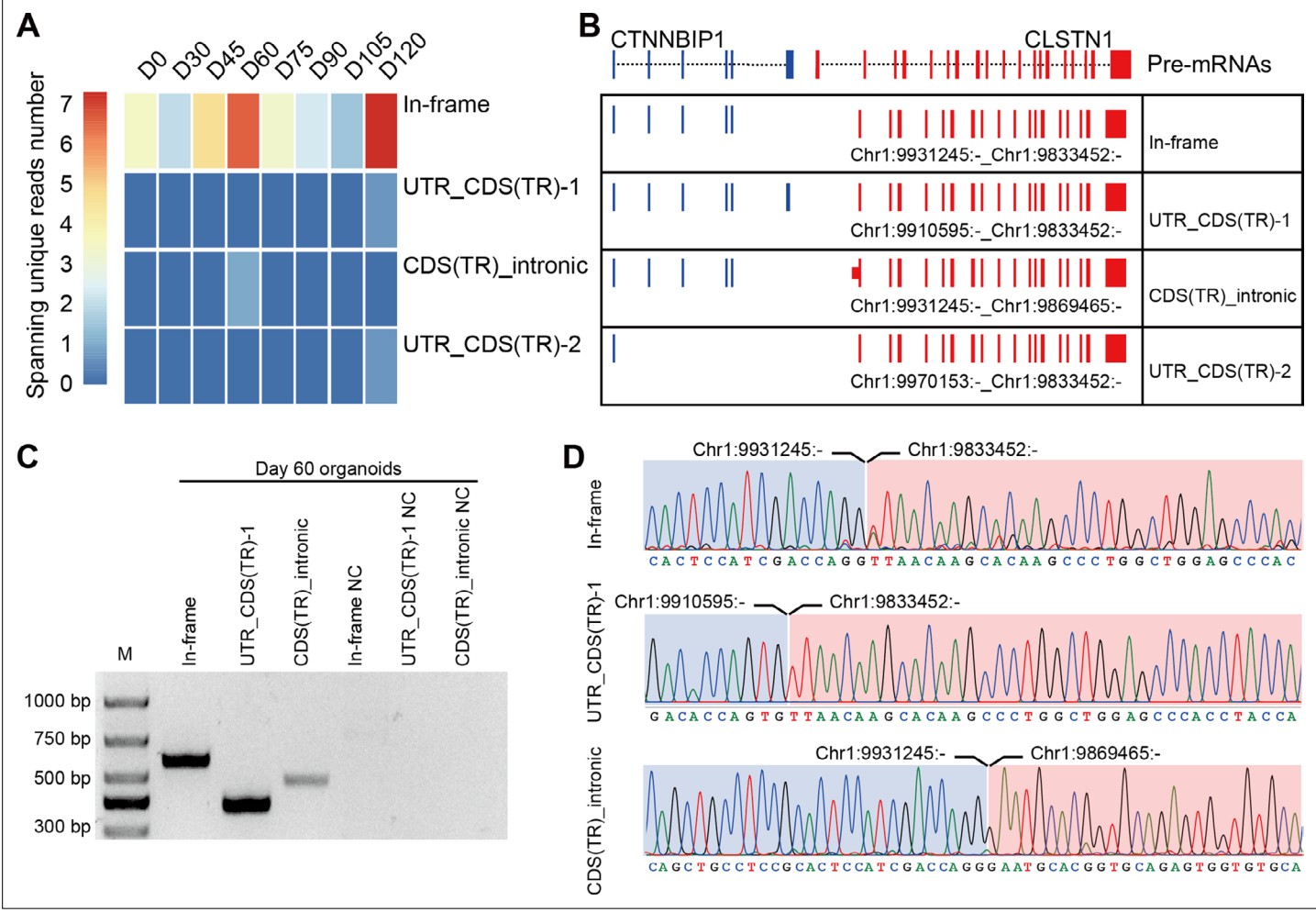

**Figure 4.** Four isoforms of *CTCL* are present in the retinal organoids (ROs). (**A**) Heatmap of *CTCL*'s spanning unique reads of each isoform in the indicated stages. UTR_CDS(TR)-1: UTR_CDS(truncated)-1; CDS(TR)Intronic: CDS(truncated)Intronic; UTR_CDS(TR)-2: UTR_CDS(truncated)-2. (**B**) Schematic diagram of the structures of the four isoforms of *CTCL*. Blue represents upstream parental gene, red represents downstream parental gene. (**C**) Four isoforms of *CTCL* in D60 ROs were validated by quantitative reverse transcription polymerase chain reaction (qRT-PCR). (**D**) Sanger sequencing to verify three isoforms of *CTCL*.

The online version of this article includes the following source data for figure 4:

**Source data 1.** Source data for *Figure 4C*.

**Source data 2.** Source data for *Figure 4C* (labelled).

uniformly positive results (*Table 1*; *Sun et al., 2019*; *Zauhar et al., 2022*). These results suggest that *CTCL* is universally generated in human retinas in vivo, ROs, and retinal cell lines in vitro, but not a phenomenon of the H9 cell line exclusively. The dynamic presence of *CTCL*, especially the In-frame isoform, suggests that it may play a vital role in human retinal development.

### *CTCL* knockdown obstructed NR differentiation but prompted the RPE differentiation

According to the chimeric *CTCL* RNA expression levels (*Figure 4A*), we focused on the D60 In-frame *CTCL* to explore its function. A CRX-tdTomato reporter hPSC line was used in the *CTCL*-knockdown experiment. This cell line is derived from H9 cells and it has been shown that tdTomato follows CRX expression spatiotemporally and the cells had the same organoid yield as H9 cells, suggesting that genetic modification of H9 did not affect its developmental process (*Pan et al., 2020*). ROs were differentiated according to the previous report (*Lowe et al., 2016*; *Figure 6—figure supplement 1* and *Video 1*). Lentiviruses carrying *shCTCL* (short-hairpin *CTCL*) or scramble *shRNA* (see Key resources

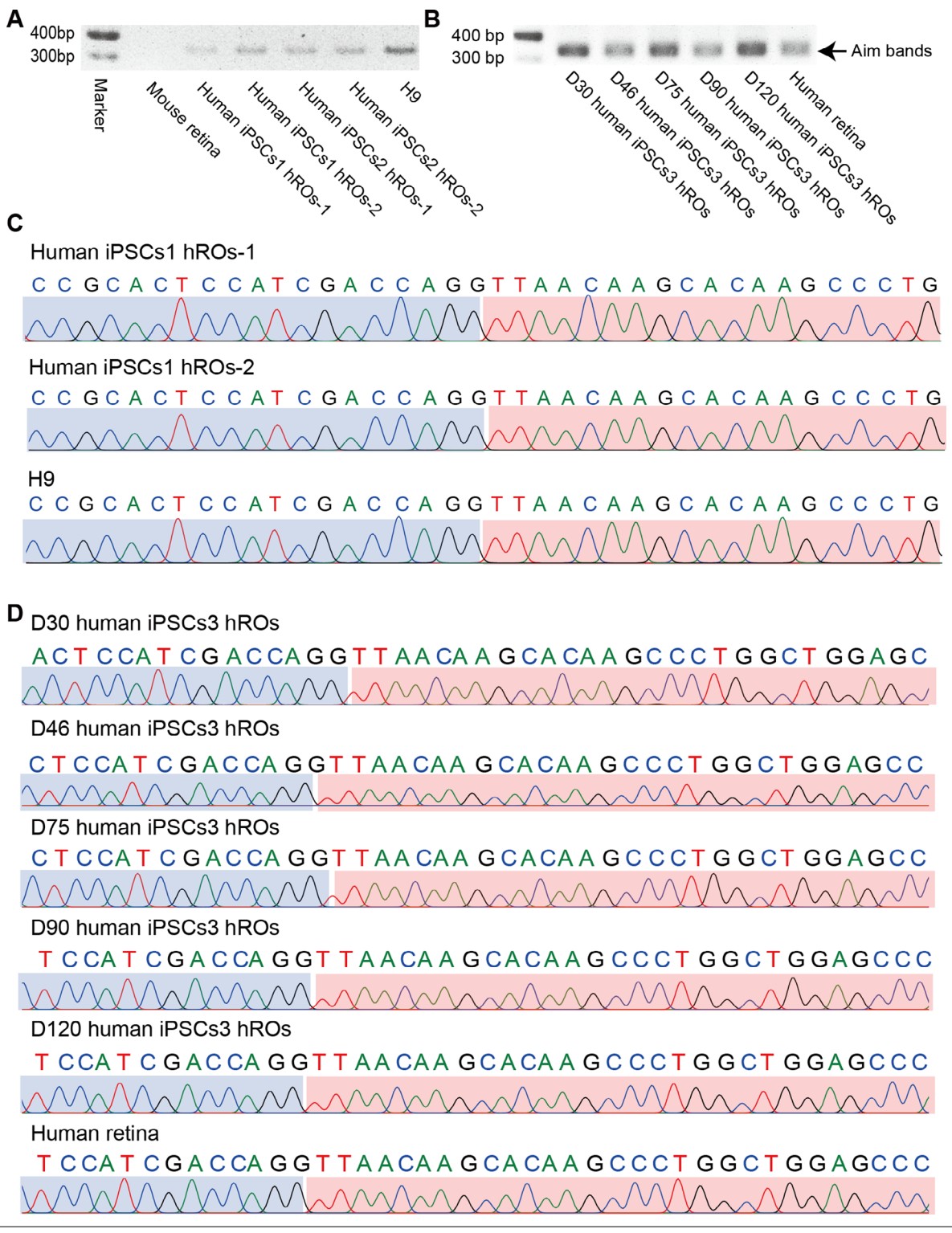

**Figure 5.** In-frame *CTCL* is common in human retinas and retinal organoids (ROs). (**A**) The qRT-PCR validation of the In-frame *CTCL* in the mouse retina, iPSCs1/iPSCs2-derived ROs, and H9 cells. (**B**) The qRT-PCR validation of In-frame *CTCL* in the iPSCs3-derived ROs and human retina. (**C**) Verification of the In-frame *CTCL* in the mouse retina, iPSCs1/iPSCs2-derived ROs and H9 cells by Sanger sequencing. (**D**) Verification of the In-frame *CTCL* in the iPSCs3-derived ROs and human retinas by Sanger sequencing.

The online version of this article includes the following source data for figure 5:

**Source data 1.** Source data for *Figure 5A*.

*Figure 5 continued on next page*

*Figure 5 continued*

**Source data 2.** Source data for *Figure 5A* (labelled).

**Source data 3.** Source data for *Figure 5B*.

**Source data 4.** Source data for *Figure 5B* (labelled).

table) infected the retinal cells at D12. After 3 days, cells were harvested for analysis of knockdown efficiency (*Figure 6A,B*). Compared with the scramble shRNA group, the In-frame *CTCL* RNA level was reduced to approximately 70% (*Figure 6B*). The RNA levels of the parental genes, *CTNNBIP1* and *CLSTN1*, showed no significant differences compared to the scramble *shRNA* group (*Figure 6—figure supplement 2*).

At D60, ROs with scramble *shRNA* displayed the typical RO morphology with the generation of photoreceptor precursors (CRX[+]) (*Figure 6C*, upper panel, live-cell imaging). In the *shCTCL* group, the organoid morphology changed with a thinner outer layer, and the CRX proteins diminished where the pigments appeared (*Figure 6C*, lower panel, live-cell imaging). The average thickness of the outer layers in those two groups was calculated (*Figure 6D*, outer layer indicated with orange arrows, live-cell imaging), and there is a dramatic decrease in the outer layer thickness in the *shCTCL* group.

To identify the aberrations in cell fate specification caused by *CTCL* knockdown, we examined the organoids with RPC-specific markers, PAX6 and OTX2. It was shown that PAX6 was highly expressed, but OTX2 was much less, and CRX appeared where GFP (*shCTC*L) was relatively lower (*Figure 6E*). CRX is known to be highly expressed in photoreceptors and precursors but low in mature RPE (*Lidgerwood et al., 2021*). Since the organoids acquired the features of RPE, we hypothesized that the outer layer of *CTCL*-knockdown organoids switched to the RPE fate. To further confirm this, the organoids were immunostained with RPE-specific markers, MITF, BEST1, and RPE65, which were present in the nucleus, cytoplasm, and intracellular regions, respectively (*Figure 6F*, *Figure 6—figure supplement 3A, C*). In contrast, no pigments or RPE-specific markers were observed in the scramble *shRNA* group, only the PAX6 was detected in the nucleus (*Figure 6—figure supplements 3B, D–F*). The *shCTCL* green fluorescence in *Figure 6E–F* was captured with live imaging and the signal was not amplified with anti-GFP antibodies; as a result, the weaker green signals can not be seen in the images. Taken together, these results indicate that *CTCL* is essential for the ROs' differentiation and that *CTCL* knockdown would promote the RPE cell fate at the expense of neural retinal cells.

**Table 1.** *CTCL* is detected in multiple healthy human retinas by RNA-seq.

| Data source | Samples | Gene 1 symbol | Gene 2 symbol | Fusion point for gene 1 | Fusion point for gene 2 |
|---|---|---|---|---|---|
| PMID: 30874468 (*Sun et al., 2019*) Data published by our lab | Human retina | CTNNBIP1 | CLSTN1 | 1:9871187:- | 1:9773394:- |
| PMID: 30874468 (*Sun et al., 2019*) Data published by our lab | Human retina | CTNNBIP1 | CLSTN1 | 1:9871187:- | 1:9773394:- |
| PMID:35784369 (*Zauhar et al., 2022*) SRR22315404 | Human retina | CTNNBIP1 | CLSTN1 | 1:9871187:- | 1:9773394:- |
| PMID:35784369 (*Zauhar et al., 2022*) SRR22315407 | Human retina | CTNNBIP1 | CLSTN1 | 1:9871187:- | 1:9773394:- |
| PMID:35784369 (*Zauhar et al., 2022*) SRR22315408 | Human retina | CTNNBIP1 | CLSTN1 | 1:9871187:- | 1:9773394:- |
| PMID:35784369 (*Zauhar et al., 2022*) SRR22315412 | Human retina | CTNNBIP1 | CLSTN1 | 1:9871187:- | 1:9773394:- |
| PMID:35784369 (*Zauhar et al., 2022*) SRR22315419 | Human retina | CTNNBIP1 | CLSTN1 | 1:9871187:- | 1:9773394:- |
| PMID:35784369 (*Zauhar et al., 2022*) SRR22315420 | Human retina | CTNNBIP1 | CLSTN1 | 1:9871187:- | 1:9773394:- |
| PMID:35784369 (*Zauhar et al., 2022*) SRR22315424 | Human retina | CTNNBIP1 | CLSTN1 | 1:9871187:- | 1:9773394:- |

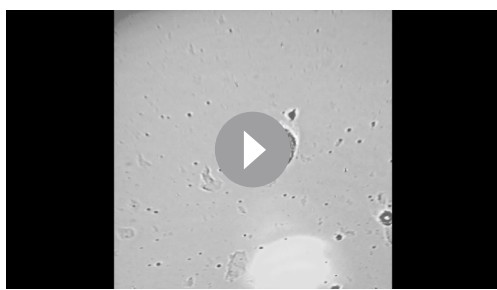

**Video 1.** The 4 days live-cell imaging of retinal organoid differentiation from D3 to D7.
https://elifesciences.org/articles/92523/figures#video1

## The underlying molecular changes in *CTCL*-knockdown ROs

The above results suggest that *CTCL* is essential for cell fate conversion in normal retinogenesis. To investigate the underlying molecular regulation, we compared the transcriptomes of *shCTCL*- and scramble *shRNA*-treated organoids by bulk RNA-seq. We obtained 3901 differentially expressed genes (DEGs) (*shCTCL* vs. scramble *shRNA*, |Log$_2$(fold change)| > 1, p-value <0.05), among which 2720 were upregulated and 1181 downregulated (*Figure 7A,B*).

To investigate the biological pathways affected by the loss-of-function of *CTCL*, we performed functional enrichment for up- and downregulated DEGs using Metascape. Upregulated DEGs were significantly enriched in pathways associated with extracellular matrix organization, morphogenesis, and epithelial cell differentiation. On the other hand, downregulated DEGs were significantly enriched in the neurogenesis-associated pathways (*Figure 7C*). To further understand the changes in cell composition between *shCTCL*- and scramble *shRNA*-treated organoids, we used the Gene Set Enrichment Analysis (GSEA) to analyze the whole-gene expression lists of the organoids. GSEA analysis with c8 as the reference gene set (*Mootha et al., 2003*; *Subramanian et al., 2005*) revealed that genes of RPE, ocular corneal, and conjunctival epithelial cells were significantly upregulated in *shCTCL* organoids; however, all genes related to neural retinal component cells (photoreceptor cells, Müller glial cells, horizontal cell, amacrine cell, and retinal ganglion cell) were downregulated (*Figure 7D*). Moreover, the expression pattern of RPE-related genes in *shCTCL* organoids was closer to that of RPE from induced pluripotent stem cells (*Maruotti et al., 2015*; *Figure 7E* and *Figure 7—figure supplement 1A*). At the same time, *CTCL* remains in a low expression state during RPE development (*Figure 7—figure supplement 1B*). These results suggest that *CTCL* has a positive role in the normal development of ROs and that *CTCL* deficiency would promote the differentiation of RPCs toward RPE.

To further explore the molecular mechanisms involved in this process, we performed the GSEA analysis using hallmark gene sets (*Mootha et al., 2003*; *Subramanian et al., 2005*). The most significantly enriched pathway was epithelial–mesenchymal transition (EMT) (*Figure 7F*). And most of the EMT-related genes were relatively highly expressed in *shCTCL*-treated organoids (*Figure 7—figure supplement 2*) (*Figure 7G*), which is particularly essential for neural crest delamination in vertebrates and the generation of different tissues during development (*Kim et al., 2014b*). Therefore, the downregulation of *CTCL* expression activated the EMT pathway which altered the directions of cell differentiation.

## Discussion

The term 'chimeric RNA' refers to any transcript composed of transcripts from different parental genes, including gene fusion transcripts (*Wu et al., 2019*). Chimeric RNAs with two parental gene sequences will further generate new dysregulated wild-type proteins, new fusion proteins, and new non-coding RNAs, expanding the abundance of the transcriptome and proteome. The first chimeric RNA that lacks DNA alterations is *AML1-ETO*, which is associated with acute myeloid leukemia (*Langabeer et al., 1997*). Subsequently, cis-splicing between neighboring genes was also found to generate chimeric RNAs, such as the CTCF-sensitive cis-spliced fusion RNAs *ADCK4-NUMBL* in prostate cancer (*Qin et al., 2015*). In addition, chimeric RNAs can be generated by long-range inter- and intra-chromosomal trans-splicing (*Li et al., 2008*). Early phase of studies has focused on the role of chimeric RNAs in cancer, where they contribute to cancer development, and serve as biomarkers and therapeutic targets.

Advances in sequencing technology have facilitated the identification of chimeric RNAs and have stimulated the research in the field of cancer. However, researchers have also found that there are some misunderstandings about chimeric RNAs. For example, *PAX3-FOXO1* was initially identified

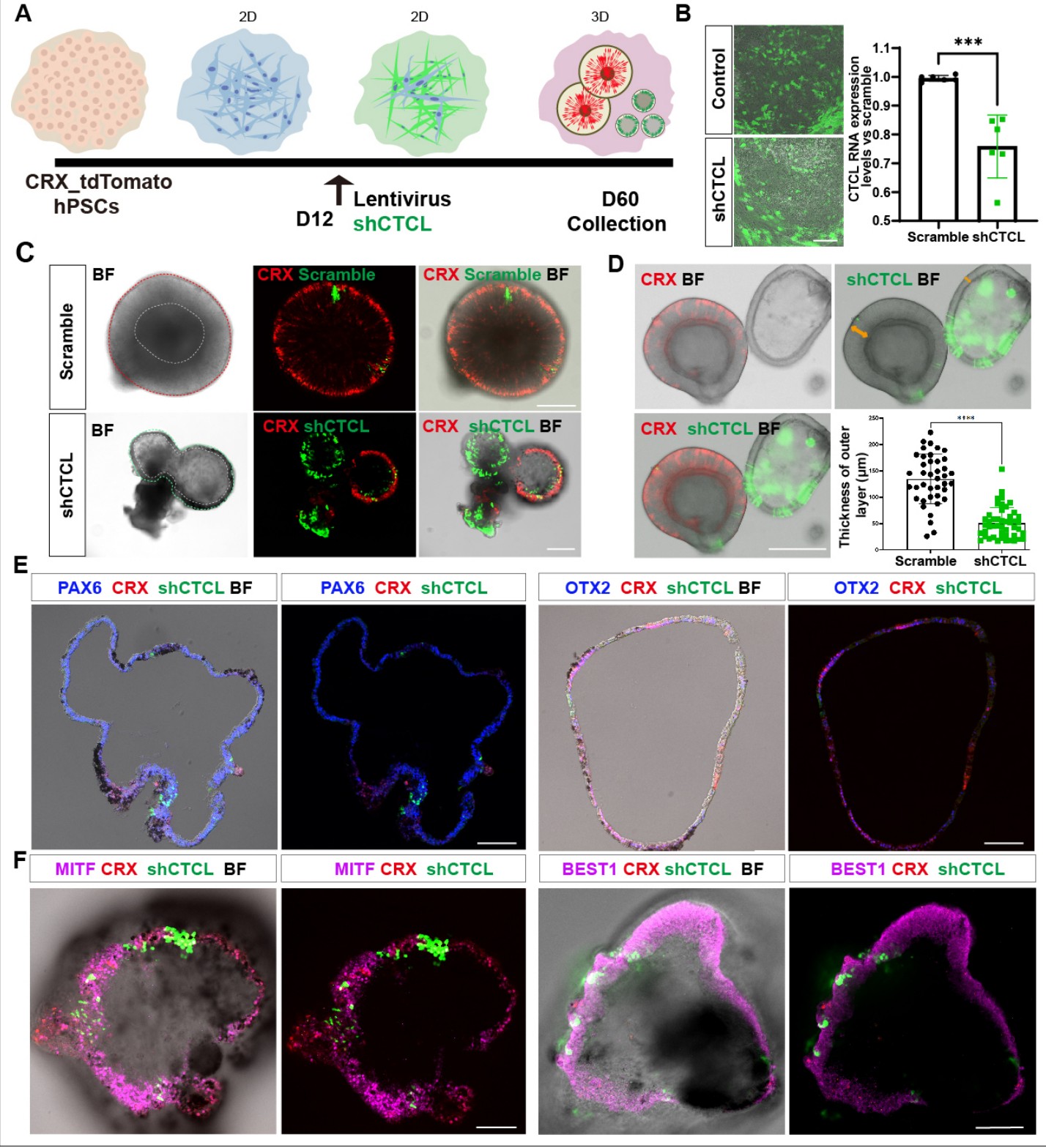

**Figure 6.** The *CTCL* knockdown obstructed neural retina's (NR) cell fates but prompted the retinal pigment epithelial (RPE) differentiation. (**A**) A schema illustrated the *shRNA* experiments. A CRX-tdTomato report line was used in this experiment. The *shCTCL* or scramble *shRNA* lentivirus transfected the retinal cells in three independent experiments on D12. All samples for *shRNA* experiments were collected on day 60, except those for analysis of knockdown efficiency. (**B**) Samples were obtained 48–72 hr after infection to examine the knockdown efficiency. There were two technical replicates for three independent experiments, *t*-test, ***, p < 0.001. Data were shown as mean ± standard deviation (SD). Scale bar = 200 μm. (**C**) D60 scramble

*Figure 6 continued on next page*

*Figure 6 continued*

*shRNA*-transfected retinal organoids (ROs) displayed a typical morphology of ROs with the expression of CRX. *ShCTCL*-transfected organoids showed thinner outer layers with much less CRX expression. BF, bright-field images, same as below. Scale bars = 200 μm. (**D**) The *shCTCL* group displayed much thinner outer layers (orange arrows in right panel). The outer layer thickness of 25 organoids from each group was measured for statistics analysis, *t*-test, ****, p < 0.0001. Data were shown as mean ± SD. Scale bar = 400 μm. (**E**) The section immunostaining of neural progenitor markers PAX6 and OTX2, and differentiated cell markers CRX, in *shCTCL*-treated organoids. Scale bars = 100 μm. (**F**) Whole-mount immunostaining of *shCTCL*-transfected organoids with CRX, *shCTCL*, and RPE-specific markers MITF and BEST1. Scale bars = 100 μm.

The online version of this article includes the following figure supplement(s) for figure 6:

**Figure supplement 1.** Retinal organoid (RO) differentiation at different timepoints.

**Figure supplement 2.** The qRT-PCR quantification of CTCL parental gene expression.

**Figure supplement 3.** Cryosection immunofluorescence staining of D60 organoids.

---

as a biomarker for alveolar rhabdomyosarcoma (*Singh et al., 2022*), but was later found to be present during normal muscle differentiation. *PAX3-FOXO1* is transiently required for muscle lineage commitment (*Xie et al., 2019*), and can escape miR-495 regulation during muscle differentiation. If *PAX3-FOXO1* is not withdrawn at the right timing, it can lead to tumorigenesis. *DUS4L-BCAP29* was originally identified in gastric cancer, where it promotes cell proliferation and is therefore considered to have tumorigenic potential (*Kim et al., 2014a*). *DUS4L-BCAP29* was subsequently identified in human non-cancerous tissues (*Tang et al., 2017*), and its role in promoting cell proliferation is not unique to cancer cells. It is a chimeric RNA present in normal physiology with a function in promoting neural differentiation (*Tang et al., 2019*). Therefore, studying chimeric RNAs in normal tissues has two implications: first, we can learn more about the normal physiological functions of chimeric RNAs; second, we can establish a control baseline for the discovery of tumor markers.

It is now well established that chimeric RNAs are abundant in normal human tissues and play a regulatory role in cellular activities. In this work, we investigated chimeric RNAs in retina for the first time. We present the expression atlas of chimeric RNAs throughout the developing ROs mimicking human retinas at various development stages. As a recently identified chimeric RNA, and the In-frame *CTCL* was reported to play a key role in cerebral development (*Ou et al., 2021*). Using the ROs as a model for loss-of-function experiments, we found that the In-frame *CTCL* also has a key role in the development of the retina and that *CTCL* deficiency inhibited NR differentiation but induced the RPE differentiation.

In-frame *CTCL* is generated by cis-splicing fusion of the first 5 exons of *CTNNBIP1* and the last 17 exons of *CLSTN1*. The 5′ parental gene *CTNNBIP1* encodes a receptor for β-catenin, and their binding promotes the catabolism of β-catenin, which in turn puts the Wnt pathway in an inactive state (*Fu et al., 2018*). Its mutation has been shown to be associated with anterior segment dysgenesis 5 in the human disease database MalaCards (https://www.malacards.org/). *CLSTN1* is a member of the calsyntenin family, a subset of the cadherin superfamily. It can mediate the axonal anterograde transport of certain types of vesicles (*Nagase et al., 1998*). The interesting question is that whether *CTCL* retains the biparental functions and/or acquires novel functions? Previous report demonstrated that *CTCL* could fine-tune Wnt signaling to regulate cerebral development (*Ou et al., 2021*). Therefore, in this study, we had expected *CTCL* to function by modulating the Wnt signaling pathway or by impacting vesicular trafficking. However, surprisingly, the GSEA and DEG functional enrichment results did not show significant changes in these two biological pathways (*Figure 6C, D, F*). The differences between the *shCTCL*- or scramble *shRNA*-treated organoids were mainly in EMT, extracellular matrix organization, morphogenesis, epithelial cell differentiation, and neurogenesis-associated pathways, suggesting that even the same chimeric RNAs may function differently in close tissues.

In summary, we have identified the chimeric RNAs in the developing human ROs and found that loss-of-function of *CTCL* inhibited NR differentiation while prompted the RPE differentiation. Although this chimeric RNA has been previously reported in the cerebral cortex and the retina is also part of the CNS, the two tissues differ in morphology, cellular composition, development, and function. Moreover, we have made new discoveries regarding the function and mechanism of *CTCL*. This study revealed for the first time a key role of chimeric RNAs in human retinal development, providing new insights into the role of chimeric RNAs in regulating tissue development. In the future, we will investigate more unreported chimeric RNAs in retinal development to further illustrate the critical role of chimeric RNAs in human retinal development.

# Materials and methods

**Key resources table**

| Reagent type (species) or resource | Designation | Source or reference | Identifiers | Additional information |
|---|---|---|---|---|
| Cell line (*Homo sapiens*) | CRX-tdTomato human ES reporter line-H9 | PMID:32831148 | | Available upon reasonable request. |
| Cell line (*Homo sapiens*) | iPSCs1 | PMID:36714839 | | Available upon reasonable request. |
| Cell line (*Homo sapiens*) | iPSCs2 | PMID:35451725 | | Available upon reasonable request. |
| Cell line (*Homo sapiens*) | iPSCs3 | PMID:33970142 | | Available upon reasonable request. |
| Antibody | anti-Pax6 (rabbit polyclonal) | Biolegend | Cat.# 862002 RRID: AB_3076431 | Dilution: 1:200 |
| Antibody | anti-Ki67 (rabbit polyclonal) | Abcam | Cat.# ab15580 RRID:AB_443209 | Dilution: 1:200 |
| Antibody | anti-OTX2　(rabbit monoclonal) | Abcam | Cat.# ab183951 RRID: AB_3076432 | Dilution: 1:200 |
| Antibody | anti-HuC/D　(mouse monoclonal) | Invitrogen | Cat.# A21271 RRID:AB_221448 | Dilution: 1:100 |
| Antibody | anti-SOX2　(mouse monoclonal) | Santa Cruz | Cat.# sc-365823 RRID:AB_10842165 | Dilution: 1:200 |
| Antibody | anti-MITF　(mouse monoclonal) | Abcam | Cat.# ab3201 RRID:AB_303601 | Dilution: 1:100 |
| Antibody | anti-GFAP (mouse monoclonal) | Santa Cruz | Cat.# sc-33673 RRID:AB_627673 | Dilution: 1:200 |
| Antibody | anti-Sox9 (rabbit mono/oligo-colonal ) | Invitrogen | Cat.# 711048 RRID:AB_2633109 | Dilution: 1:200 |
| Antibody | anti- RxRγ (mouse monoclonal) | Santa Cruz | Cat.# sc-365252 RRID:AB_10850062 | Dilution: 1:100 |
| Recombinant DNA reagent | pLenti-U6-shRNA-EF1a-EGFP-T2A-Puro-WPRE | This paper | | Available upon reasonable request. |
| Sequence-based reagent | UTR_CDS(truncated)-1 CTCL-forward | Tsingke Biotechnology Co., Ltd | | TGCAAAGCCCTTGGAACA |
| Sequence-based reagent | UTR_CDS(truncated)-1 CTCL-reverse | Tsingke Biotechnology Co., Ltd | | TCCACTACCACTGCATCAAAG |
| Sequence-based reagent | in-frame CTCL-forward | Tsingke Biotechnology Co., Ltd | | TTCCTACTTCTGCCCAGCC |
| Sequence-based reagent | in-frame CTCL-reverse | Tsingke Biotechnology Co., Ltd | | AGGCCTGGATGGTGAATGAAT |
| Sequence-based reagent | CDS(truncated)_intronic CTCL-forward | Tsingke Biotechnology Co., Ltd | | GAAGAGTCCGGAGGAGATGTA |
| Sequence-based reagent | CDS(truncated)_intronic CTCL-reverse | Tsingke Biotechnology Co., Ltd | | ATTCTCTGCCAAGACTTACACC |
| Sequence-based reagent | UTR_CDS(truncated)-2 CTCL-forward | Tsingke Biotechnology Co., Ltd | | CTCCTGCTGCTGCTACTG |

*Continued on next page*

*Continued*

| Reagent type (species) or resource | Designation | Source or reference | Identifiers | Additional information |
|---|---|---|---|---|
| Sequence-based reagent | UTR_CDS(truncated)-2 CTCL-reverse | Tsingke Biotechnology Co., Ltd | | GGTCCCTTCCCACAATCATAG |
| Sequence-based reagent | GAPDH forward | Tsingke Biotechnology Co., Ltd | | CTCTGACTTCAACAGCGACA |
| Sequence-based reagent | GAPDH reverse | Tsingke Biotechnology Co., Ltd | | GTAGCCAAATTCGTTGTCATACC |
| Sequence-based reagent | ShCTCL | Tsingke Biotechnology Co., Ltd | | TGCTTGTTAACCTGGTCGA |
| Sequence-based reagent | Scramble | Tsingke Biotechnology Co, Ltd | | GCCTAAGGTTAAGTCGCCCTCG |
| Software, algorithm | R (v4.0.3) | *R Development Core Team, 2020* | RRID:SCR_001905 | https://www.r-project.org/ |
| Software, algorithm | FusionCatcher | *Nicorici, 2023* | | https://github.com/ndaniel/fusioncatcher |
| Software, algorithm | Hisat2 | *Kim and Park, 2022* | | https://github.com/DaehwanKimLab/hisat2 |
| Software, algorithm | samtools | *Danecek et al., 2023* | | https://github.com/samtools/samtools |
| Software, algorithm | featureCounts | *Liao et al., 2021* | | http://subread.sourceforge.net/featureCounts.html |
| Software, algorithm | ggplot2 | *Pedersen, 2022* | | https://rdocumentation.org/packages/ggplot2/versions/3.3.6 |
| Software, algorithm | ggseqlogo | *Wagih, 2017* | | https://github.com/omarwagih/ggseqlogo |
| Software, algorithm | chimeraviz | *Lågstad, 2023* | | https://www.bioconductor.org/packages/release/bioc/html/chimeraviz.html |
| Software, algorithm | pheatmap | *Kolde, 2019* | | https://rdocumentation.org/search?q=pheatmap |
| Software, algorithm | clusterProfiler | *Yu, 2012* | | https://rdocumentation.org/search?q=clusterProfiler |

## RNA sequencing and data analysis

A total amount of 1–3 µg RNA per sample was used as input material for the RNA sample preparations. Sequencing libraries were generated using VAHTS Universal V6 RNA-seq Library Prep Kit for Illumina (NR604-01/02) following the manufacturer's recommendations and index codes were added to attribute sequences to each sample. Briefly, mRNA was purified from total RNA using poly-T oligo-attached magnetic beads. Then we added fragmentation buffer to break the mRNA into short fragments. First-strand cDNA was synthesized using random hexamer primer and RNase H. Second-strand cDNA synthesis was subsequently performed using buffer, dNTPs, DNA polymerase I, and RNase H. And then, the double stranded cDNA was purified by AMPure P beads or QiaQuick PCR kit. The purified double-stranded cDNA was repaired at the end, added a tail and connected to the sequencing connector, then the fragment size was selected, and finally the final cDNA library was obtained by PCR enrichment.

We used FusionCatcher (*Nicorici et al., 2014*) software (https://github.com/ndaniel/fusioncatcher) to identify chimeric RNAs in human ROs. Positive chimeric RNAs identified using FusionCatcher were selected with alignment of spanning unique reads. The expression level of chimeric RNAs was reflected from log(Spanning_unique_reads/Total_number_of_reads*10,000,000) value. To analyze the expression of parental genes in human ROs, raw reads were first mapped to the hg38 human genome reference sequence by Hisat2 software, then transcripts were assembled with featureCounts. Metascape

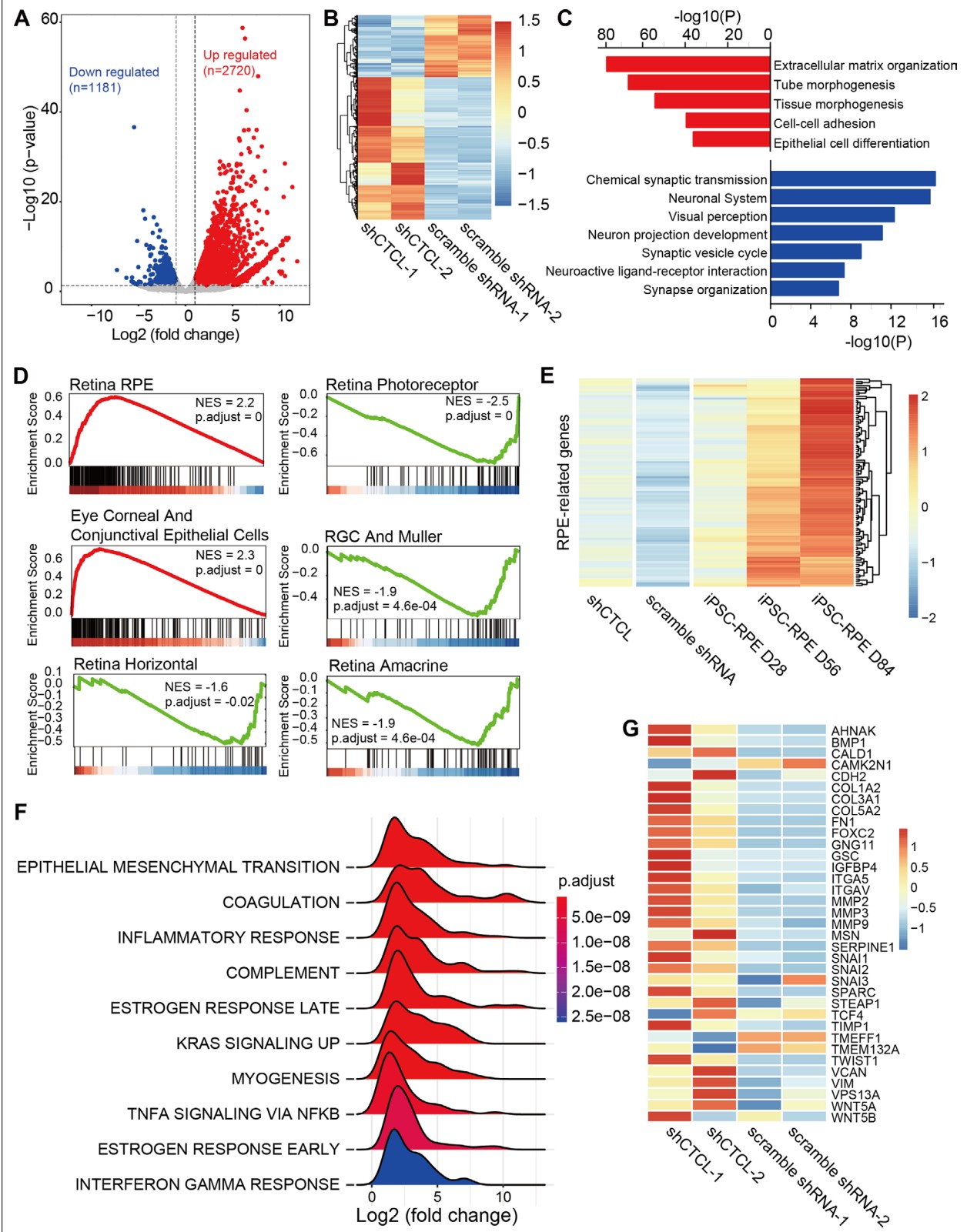

**Figure 7.** Transcriptomic alterations in *CTCL*-knockdown retinal organoids. (**A**) Volcano plot shows differentially expressed genes (DEGs, p-values <0.05 and |Log$_2$(*shCTCL*/scramble)| > 1) between *shCTCL*- and scramble *shRNA*-treated organoids at D60. (**B**) Heatmap of DEGs in *shCTCL*- and scramble *shRNA*-treated organoids. (**C**) Functional enrichment analysis of up- and downregulated DEGs. (**D**) Gene Set Enrichment Analysis (GSEA) results showed the enriched gene sets in *shCTCL*-treated organoids using the c8 reference gene set. (**E**) Expression of retinal pigment epithelial (RPE)-related genes

*Figure 7 continued on next page*

*Figure 7 continued*

in *shCTCL*- and scramble *shRNA*-treated organoids. (**F**) GSEA results showed the enriched gene sets in *shCTCL*-treated organoids using the hallmark reference gene set. (**G**) Expression of epithelial–mesenchymal transition (EMT)-related genes in *shCTCL*- and scramble *shRNA*-treated organoids.

The online version of this article includes the following figure supplement(s) for figure 7:

**Figure supplement 1.** Expression of CTCL is lower in differentiated retinal pigment epithelial (RPE) cells.

**Figure supplement 2.** Validation of epithelial–mesenchymal transition (EMT)-related genes by PCR.

(https://metascape.org/gp/index.html) is used for functional annotation. The position weight matrix of the 20 bp DNA sequence motif around the fusion site was calculated by the seqLogo R package. GSEA analysis is done with the R package clusterProfiler.

## Generation of ROs from hESCs

The CRX-tdTomato human ES reporter line and iPSCs were used for the ROs differentiation. The hESCs were cultured to 80% confluence. Cell colonies were dissociated using dispase buffer (Stem cell, 07923) for 5 min at 37°C and were then cut into smaller pieces/aggregates. The aggregates were collected and mixed with Matrigel (RD). The solidified gel with ES cells were dispersed into small pieces with medium containing 1:1 mixture of Dulbecco's modified Eagle medium (DMEM)/F12, neurobasal medium, and 0.5× N2 supplement (Gibco), 0.5× B27 supplement (Gibco), 1× MEM Non-Essential Amino Acids Solution (Gibco), 2 mM Glutamax (Gibco), and 0.1 mM 2-mercaptoenthanol. At day 1 (D1), hollow cysts could be observed in the dish and cysts started to attach to the culture dish and spread in 3 days. At D5, the cysts were dispersed into two culture dishes with same medium. Medium was changed every 5 days. On D15, detaching the cells with dispase and change to medium with DMEM/F12 (3:1), 1× B27 supplement (Gibco), and 1× MEM-NEAA (Gibco) for a week. Optic vesicles formed in this period. Finally, optic vesicles were transfered to serum medium with DMEM/F12 (3:1), 1× B27 supplement (Gibco), 1× MEM-NEAA (Gibco), 8% fetal bovine serum (Gibco), 100 mM Taurine (Sigma-Aldrich), and 2 mM Glutamax (Gibco). ROs would be collected at D60.

## *CTCL* knockdown in the ROs

RNA interference oligo (5'-TGCTTGTTAACCTGGTCGA-3') against *CTCL* was cloned into lentivirus vector (pLenti-U6-shRNA-EF1a-EGFP-T2A-Puro-WPRE). D12 retinal cells were infected with lentiviruses ( Multiplicity of infection [MOI] = 10, the ratio of the number of transducing lentiviral particles to the number of cells) for 6 hr, and replaced with fresh medium. The fluorescence was detected after 48 hr. The expression of *CTCL*, *CTNNBIP1*, and *CLSTN1* in retinal cells was analyzed at D15 by quantitative PCR.

## Immunostaining of cryosections

ROs were fixed in 4% paraformaldehyde for 30 min and imbedded in O.C.T. compound and sectioned into 10 μm slices. The cryosections were blocked with 0.5% Triton X-100 in 4% bovine serum albumin (BSA) for 1 hr. After that, sections were incubated in primary antibodies (diluted in 4% BSA supplied with 0.5% Triton X-100) at 4°C overnight. The following primary antibodies were used: anti-OTX2 (1:200, Cat.# ab183951; Abcam), anti-PAX6 (1:200, Cat.# 901301; Biolegend), anti-SOX2 (1:200, Cat.# sc-365823; Santa Cruz), anti-HuC/D (1:100, Cat.# A21271; Invitrogen), anti-MITF (1:100, Cat.# ab3201; Abcam), anti-GFAP (1:200, Cat.# sc-33673; Santa Cruz), anti-Sox9 (1:200, Cat.# 711048; Invitrogen), anti-RxRγ (1:100, Cat.# sc-365252; Santa Cruz), and anti-Ki67 (1:200, Cat.# ab15580; Abcam). After washed with phosphate-buffered saline, cryosections were stained with Alexa Fluor-conjugated secondary antibodies (diluted 1:500, Invitrogen) for 1 hr at room temperature in the dark.

## Quantification and statistical analysis

Heatmap was plotted with the R package Pheatmap. All plots were drawn with R package ggplot2. Detailed statistical analysis of experiments can be found in the figure legends, including the statistical tests used, exact values, and biological replicates.

## Resource availability

Further information and requests for resources and reagents should be directed to and will be fulfilled by the lead contact, Professor Zi-Bing Jin (jinzb502@ccmu.edu.cn).

## Materials availability

All unique/stable reagents generated in this study are available from the lead contact with a completed materials transfer agreement.

## Acknowledgements

This study was partially supported by grants from National Natural Science Foundation of China (82125007).

## Additional information

### Funding

| Funder | Grant reference number | Author |
|---|---|---|
| National Natural Science Foundation of China | 82125007 | Zi-Bing Jin |

The funders had no role in study design, data collection, and interpretation, or the decision to submit the work for publication.

### Author contributions

Wen Wang, Conceptualization, Data curation, Software, Formal analysis, Validation, Investigation, Visualization, Methodology, Writing - original draft, Writing – review and editing; Xiao Zhang, Formal analysis, Validation, Investigation, Visualization, Methodology; Ning Zhao, Ze-Hua Xu, Investigation; Kangxin Jin, Writing – review and editing; Zi-Bing Jin, Conceptualization, Resources, Supervision, Funding acquisition, Project administration, Writing – review and editing

### Author ORCIDs

Wen Wang ⓘ http://orcid.org/0000-0002-8601-7385
Xiao Zhang ⓘ http://orcid.org/0000-0002-8599-3181
Kangxin Jin ⓘ https://orcid.org/0000-0002-0108-6948
Zi-Bing Jin ⓘ https://orcid.org/0000-0003-0515-698X

### Ethics

One human donor retinal sample was included in this study. The study was approved by the Ethics Committee of Beijing Tongren Hospital (No. TRECKY2021-089) and conducted in accordance with the Declaration of Helsinki.

### Decision letter and Author response

Decision letter https://doi.org/10.7554/eLife.92523.sa1
Author response https://doi.org/10.7554/eLife.92523.sa2

## Additional files

### Supplementary files

• MDAR checklist

### Data availability

Sequencing data has been deposited in GSA under accession code HRA005688 and may be accessed by application to a data access committee. The Gene Expression Omnibus (GEO) number for the bulk RNA-seq of organoids in this paper is GSE136929. All data generated or analyzed during this study is included in the manuscript and supporting files. Source data files have been provided for Figures 4C and 5A,B.

The following dataset was generated:

| Author(s) | Year | Dataset title | Dataset URL | Database and Identifier |
| --- | --- | --- | --- | --- |
| Jin XB | 2024 | CTCL in RO | https://ngdc.cncb.ac.cn/gsa-human/browse/HRA005688 | National Genomics Data Center, HRA005688 |

The following previously published datasets were used:

| Author(s) | Year | Dataset title | Dataset URL | Database and Identifier |
| --- | --- | --- | --- | --- |
| Liu H, Zhang Y, Zhang Y, Li Y, Hua Z, Han F, Jin S, Zhang C, Wu K, Lv J, Yu F, Lin Q, Chen C, Su J, Jin Z | 2020 | Human embryonic stem cell-derived organoidal retinoblastoma reveals cancerous origin and therapeutic target | https://www.ncbi.nlm.nih.gov/geo/query/acc.cgi?acc=GSE136929 | NCBI Gene Expression Omnibus, GSE136929 |
| Stambolian D | 2022 | Human AMD | https://www.ncbi.nlm.nih.gov/geo/query/acc.cgi?acc=GSE155154 | NCBI Gene Expression Omnibus, GSE155154 |

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
