## [Editor Report]

The primary goal of this paper is to profile expression of chimeric RNAs in human retinal organoids throughout development. The fundamental findings provide an important step forward in understanding retinogenesis specifically and neurogenesis, generally. The compelling evidence unveils a previously unrecognized significant role of chimeric RNA CTCL in human retinal development.

---

## [Decision Letter]

**Decision letter after peer review:**

Thank you for submitting the paper "Profiling of chimeric RNAs in human retinal development with retinal organoids" for consideration by *eLife*. Your article has been reviewed by 3 peer reviewers, one of whom is a member of our Board of Reviewing Editors, and the evaluation has been overseen by a Senior Editor.

Comments to the Authors:

We are sorry to say that, after consultation with the reviewers, we have decided that this work will not be considered further for publication by *eLife*.

The consensus of the reviewers is that, while the work has strong potential for impact, concerns surrounding replication and reproducibility undermine the findings in this manuscript. Below is a detailed summary of reviewer comments:

Strengths

1. Novel study from the perspective of defining a role for chRNAs in early retinal development.

2. Criteria and characterization parameters for the chimeric dataset are well-described and thoughtful.

3. Evaluation of CTCL provides a functional valuation of the chimeric dataset and identifies a potential role in epithelial-mesenchymal transition (EMT), which is an interesting mechanism for cell fate switches in developmental and pathological processes.

Weaknesses

1. Studies appear to be limited to the use of a single cell line, which does not take into account variability between cell lines or between individuals. The field standard is 3 parent lines.

2. While the coverage of 8 developmental stages is commendable, it seems that this suggests that an average of less than 3 retinal organoids was assessed at each developmental stage. With significant variability often observed between organoids, the use of an extremely limited number of organoids like this is further concerning.

3. While the authors have published on the differentiation of retinal organoids in the past, and they discuss some features of retinal organoids at various developmental stages in this manuscript, it would strengthen the paper for the casual reader if some of this info was presented in Figure 1. For example, what is the significance of each developmental stage assessed, and how are the organoids characterized at each point? Representative immunostaining of organoids along with a description at each time point would greatly assist in the understanding of this study for those that are not up to date on retinal organoids.

4. Concerns exist about the ability to reproduce these studies based on methods provided in the manuscript. For example, while the authors state that the "CRX-tdTomato human ES reporter line was used for the ROs differentiation", this does not state which cell line this actually is, nor whether this is an established cell line that has been previously characterized or not. More detail needs to be provided here and if this is a new reporter line extensive characterization needs to be performed.

5. Some representative images in Figure 5 suggest that most cells are not transduced with shRNA, and sometimes very few are transduced. This is particularly concerning for scramble controls such as those observed in Figure 5C, where it appears that perhaps just 1 or 2 cells are transduced based upon the GFP reporter. It is not clear how much this variability affects these experiments, and the figures as presented suggest that scramble controls may not be better than untransduced cultures.

6. Evidence for EMT is solely based on RNA sequencing. The authors should at least provide the qRT-PCR verifications.

7. The Discussion section should provide greater context for the findings of the study, including recent progress in the roles of chRNAs (and the mechanisms) in tissue development and maintenance, differential functions in tumors and normal tissues should have compared and distinguished in the discussion, and relevance of loss of function studies to the retina, i.e. anterior segment dysgenesis 5 is a developmental disorder of the eye associated with mutations in CTNNBIP1.

*Reviewer #1 (Recommendations for the authors):*

The purpose of this study is to profile chimeric RNAs expression during human retinal development. Chimeric RNAs regulate stem/progenitor cell differentiation, stemness maintenance and central nervous system development. Using human embryonic stem cells (hESC)-derived retinal organoids (day 0 to day 120), the study profiled chimeric RNAs with RNA sequencing. This profiling identified several chimeric RNAs with commonality to neuronal differentiation and stem cell maintenance as well as chimeric RNAs that appear specific to retinogenesis. Based on these data, the functional significance of chimeric RNA CTNNBIP1-CLSTN1 (CTCL) in retinogenesis was examined. CTCL shRNA studies in human retinal organoids revealed that CTCL loss-of-function obstructs neural differentiation, but promotes of retinal pigment epithelial differentiation. These data provide an important step forward in understanding retinogenesis specifically and central nervous system development, generally.

Strengths

1. Reports first, the unbiased profile of chimeric RNA expression across a broad developmental period in human-derived retinal organoids.

2. Criteria and characterization parameters for the chimeric dataset are well-described and thoughtful.

3. Functional follow-up on CTCL identifies a novel role for this chimeric RNA in human retinogenesis and demonstrates the utility of the chimeric RNA profile more broadly.

4. The functional consequence of CTCL downregulation has novel implications for developmental disorders of the retina.

Weaknesses

1. While justified by realistic and pervasive limitations, all studies here are performed in human retinal organoids. The relevance of this dataset to human development in vivo is unknown. As such, all findings from this study should be interpreted with the caveat that they may represent the characterization of a humanoid model that is not directly translatable to human development and disease.

1. In the Discussion, it would be beneficial to mention Anterior segment dysgenesis 5, which is a developmental disorder of the eye associated with mutations in CTNNBIP1. This disorder is one of a collection of anterior segment dysgeneses that affect the entire anterior segment of the eye, including the cornea, iris, lens, trabecular meshwork, and Schlemm canal. This could provide more insight and context for the findings of the loss-of-function of studies.

*Reviewer #2 (Recommendations for the authors):*

In the manuscript by Wang et al., the authors reported the discovery of the chimeric RNAs (chRNAs), their quantities, and categories in human retinae. Using human iPS-derived retinal organoids as a model system, they described the dynamic expression patterns of the chRNAs during development and identified certain chRNAs that are constantly expressed during the course of development studied (up to 120 days). The function of chRNAs, taking CTCL as an example, was further demonstrated by loss-of-function experiments. They demonstrated that the knockdown of CTCL led to a reduced photoreceptor population while more cells expressing RPE-specific genes are observed. Lastly, they showed that the epithelium-mesenchymal transition is the underlying mechanism for such a fate switch from photoreceptor to RPE cells. This is the first study of the role of chimeric RNAs in early retinal development. The usage of human iPS-derived retinal organoids is proper for this subject. The paper is concisely written and should be of interest to the field of developmental biology and vision research.

1) The paper for the first time revealed the landscape of chRNAs in the retina during early development. 2) The paper demonstrated that chRNAs may have important functions during neural tissue development. 3) The function of chRNAs may be tissue specific. 3) The potential mechanism of CTCL in epithelial-mesenchymal transition (EMT), an interesting mechanism involved in cell fate switches in developmental and pathological processes.

Main concerns: the evidence for EMT is solely based on RNA sequencing. The authors should at least provide the qRT-PCR verifications. A few newest pieces of literature should be cited. In the Discussion section, recent progresses in the roles of chRNAs (and the mechanisms) in tissue development and maintenance should be addressed more extensively. Their differential function in tumors and normal tissues should have been compared and distinguished in the discussion.

*Reviewer #3 (Recommendations for the authors):*

In the manuscript by Wang et al., the authors set out to profile the expression of chimeric RNAs within the retina, using human pluripotent stem cell-derived retinal organoids as the model. Chimeric RNAs are typically characterized by the inclusion of exons from different genes and as such, they have not been as extensively studied as other gene targets. However, studies have demonstrated an important role for some chimeric RNAs in both normal development as well as in some disease states such as tumorigenesis. Despite this, relatively little is known about the expression patterns of chimeric RNAs in development, including within the development of the retina. To this end, the authors have pursued approaches to address this shortcoming by using human retinal organoid models. In the study, the authors first characterize the expression of chimeric RNAs in retinal organoids at various stages of development, from undifferentiated stem cells until 120 days of differentiation, at which point many (but not all) retinal cell types are represented. To then further demonstrate the significance of these chimeric RNAs, the authors then focus their studies upon one in particular – CTCL and assess the function of this chimeric RNA in loss-of-function studies that suggest that the early expression of CTCL in retinal development predisposes tissue toward a neural retina fate rather than RPE. Taken together, the studies are presented logically and are of interest to the scientific community, but a number of issues were identified that should be addressed to assist in the clarity of the research presented as well as the reproducibility of these studies for others.

First, while the authors have published on the differentiation of retinal organoids in the past, and they discuss some features of retinal organoids at various developmental stages in this manuscript, it would strengthen the paper for the casual reader if some of this info was presented in Figure 1. For example, what is the significance of each developmental stage assessed, and how are the organoids characterized at each point? Representative immunostaining of organoids along with a description at each timepoint would greatly assist in the understanding of this study for those that are not up to date on retinal organoids.

Next, some concerns exist about the ability to reproduce these studies based on methods provided in the manuscript. For example, while the authors state that the "CRX-tdTomato human ES reporter line was used for the ROs differentiation", this does not state which cell line this actually is, nor whether this is an established cell line that has been previously characterized or not. More detail needs to be provided here and if this is a new reporter line extensive characterization needs to be performed. Further, it seems that the studies were limited to the use of a single cell line, which does not take into account variability between cell lines nor between individuals, further reducing the potential impact of these results.

Next, it seems concerning that the authors "examined chimeric RNAs in 22 human ROs covering eight developmental stages". While the coverage of 8 developmental stages is commendable, it seems that this suggests that an average of less than 3 retinal organoids was assessed at each developmental stage. With significant variability often observed between organoids, the use of an extremely limited number of organoids like this is further concerning, as it may not truly represent what may be found in other studies.

Some concern also exists in the experimental manipulation of CTCL in the latter parts of the paper. The authors have selected this chimeric RNA as a proof of principle based on its observed expression in their studies, as well as previous studies that have assessed this RNA in cerebral organoids. To manipulate CTCL expression the authors have used shRNA approaches to knock down expression, in which their studies demonstrate an approximately 25% reduction in expression. However, some representative images in Figure 5 suggest that most cells are not transduced, and sometimes very few are transduced. This is particularly concerning for scramble controls such as those observed in Figure 5C, where it appears that perhaps just 1 or 2 cells are transduced based upon the GFP reporter. It is not clear how much this variability affects these experiments, and the figures as presented suggest that scramble controls may not be better than untransduced cultures.

Finally, given the uncertainty around the limited numbers of organoids assessed (at least as determined by the studies as described by the authors) as well as the apparent difficulty in reproducibly transducing the organoids, it is concerning that the results presented will not be reproducible by others. While the studies are generally of interest, these shortcomings represent a concern.

To improve the manuscript, I recommend that the authors address multiple areas to improve reproducibility across studies. Either sample sizes need to be increased or if an error has been made in this review, then more effort needs to be made to describe sample sizes in the manuscript. Additional cell lines should also be used to validate these findings across subjects.

Further, I suggest justification of the studies further to create a more impactful, cohesive story. The study as it currently exists is a straightforward profiling of chimeric RNAs in the first few figures in a descriptive manner, and then an assessment of CTCL in the latter half. However, the selection of CTCL is not really based on the profiling and represents a somewhat incremental advance over previous studies in cerebral organoids. For justification in *eLife*, I would recommend further efforts made to integrate new findings of chimeric RNAs in retinal organoids.

---

## [Author Response]

Comments to the Authors:Weaknesses1. Studies appear to be limited to the use of a single cell line, which does not take into account variability between cell lines or between individuals. The field standard is 3 parent lines.

We thank the reviewers for providing helpful suggestions which make the conclusions more solid. To prove that *CTCL* is not H9 cell line-specific, we assayed *CTCL* in naive human retinal tissues, retinal organoids cultured from five cell lines (3 human iPSCs and 2 human ESCs). Results of PCR and Sanger sequencing showed that *CTCL* was present in these tissues (Figure 5). More importantly, we also utilized human retinal transcriptomic sequencing data available in the GEO database to detect *CTCL* chimeric RNA and obtained positive results (Table 1). All of these results confirm that *CTCL* is not specific to the H9 cell line and that it is widely present in human retinas, organoids and cell lines.

2. While the coverage of 8 developmental stages is commendable, it seems that this suggests that an average of less than 3 retinal organoids was assessed at each developmental stage. With significant variability often observed between organoids, the use of an extremely limited number of organoids like this is further concerning.

We thank the reviewers for raising this concern. We actually collected a total of 22 retinal organoid samples, each containing approximately 100 organoids, and that these 22 samples covered 8 developmental stages with 2-3 biological replicates per developmental stages. We revised the context to make this clearer.

3. While the authors have published on the differentiation of retinal organoids in the past, and they discuss some features of retinal organoids at various developmental stages in this manuscript, it would strengthen the paper for the casual reader if some of this info was presented in Figure 1. For example, what is the significance of each developmental stage assessed, and how are the organoids characterized at each point? Representative immunostaining of organoids along with a description at each time point would greatly assist in the understanding of this study for those that are not up to date on retinal organoids.

We thank the reviewers for pointing this out. We have added relevant content to the manuscript (page5, line 96-103): On day 0 (D0) of differentiation, hESCs in a good growth state were dissociated into individual cells and seeded at a certain cell density in V-shaped low-adhesion culture dishes so that the cells would aggregate into clusters to induce the neuroepithelial formation. At D30 of differentiation, the RO outermost layer with translucent NR structures was visible. As the differentiation time prolonged, the NR gradually became denser. At D105 of differentiation, ROs could be seen with the formation of RPE. At D120 of differentiation, the outer segments of photoreceptor cells were seen on the surface of ROs.

4. Concerns exist about the ability to reproduce these studies based on methods provided in the manuscript. For example, while the authors state that the "CRX-tdTomato human ES reporter line was used for the ROs differentiation", this does not state which cell line this actually is, nor whether this is an established cell line that has been previously characterized or not. More detail needs to be provided here and if this is a new reporter line extensive characterization needs to be performed.

We agree and have updated. We have added a description of the cell line to the manuscript (page20, line 220-224) A CRX-tdTomato reporter hPSC line was used in the *CTCL* knockdown experiment. This cell line is derived from H9 cells and it has been shown that tdTomato follows CRX expression spatiotemporally and the cells had the same organoid yield as H9 cells, suggesting that genetic modification of H9 did not affect its developmental process (Pan et al., 2020).

Pan, D., Xia, X. X., Zhou, H., Jin, S. Q., Lu, Y. Y., Liu, H.,... Jin, Z. B. (2020). COCO enhances the efficiency of photoreceptor precursor differentiation in early human embryonic stem cell-derived retinal organoids. Stem Cell Res Ther, 11(1), 366. https://doi.org/10.1186/s13287-020-01883-5

5. Some representative images in Figure 5 suggest that most cells are not transduced with shRNA, and sometimes very few are transduced. This is particularly concerning for scramble controls such as those observed in Figure 5C, where it appears that perhaps just 1 or 2 cells are transduced based upon the GFP reporter. It is not clear how much this variability affects these experiments, and the figures as presented suggest that scramble controls may not be better than untransduced cultures.

We thank the reviewers for this helpful comment. The green fluorescence in the images is captured from the live cell imaging and NOT enhanced with anti-eGFP antibodies, and as a result, the weaker green signals could not be seen in these images, but these green signals are actually visible under fluorescence microscope.

6. Evidence for EMT is solely based on RNA sequencing. The authors should at least provide the qRT-PCR verifications.

We performed qRT-PCR analysis of the EMT genes and the result is consistent with the RNA-seq. In the manuscript we put this result in Figure 7—figure supplement 2.

7. The Discussion section should provide greater context for the findings of the study, including recent progress in the roles of chRNAs (and the mechanisms) in tissue development and maintenance, differential functions in tumors and normal tissues should have compared and distinguished in the discussion, and relevance of loss of function studies to the retina, i.e. anterior segment dysgenesis 5 is a developmental disorder of the eye associated with mutations in CTNNBIP1.

Thank you for your advice. We have made such changes. The new sentence reads as follows “Advances in sequencing technology have facilitated the identification of chimeric RNAs and have stimulated the research in the field of cancer. However, researchers have also found that there are some misunderstandings about chimeric RNAs. For example, *PAX3-FOXO1* was initially identified as a biomarker for alveolar rhabdomyosarcoma (Singh et al., 2022), but was later found to be present during normal muscle differentiation. *PAX3-FOXO1* is transiently required for muscle lineage commitment (Xie et al., 2019), and can escape miR-495 regulation during muscle differentiation. If *PAX3-FOXO1* is not withdrawn at the right timing, it can lead to tumorigenesis. *DUS4L-BCAP29* was originally identified in gastric cancer, where it promotes cell proliferation and is therefore considered to have tumorigenic potential (H. P. Kim et al., 2014). *DUS4L-BCAP29* was subsequently identified in human non-cancerous tissues (Tang et al., 2017), and its role in promoting cell proliferation is not unique to cancer cells. It is a chimeric RNA present in normal physiology with a function in promoting neural differentiation (Tang et al., 2019). Therefore, studying chimeric RNAs in normal tissues has two implications: first, we can learn more about the normal physiological functions of chimeric RNAs; second, we can establish a control baseline for the discovery of tumor markers” and “In-frame *CTCL* is generated by cis-splicing fusion of the first 5 exons of *CTNNBIP1* and the last 17 exons of *CLSTN1*. The 5' parental gene *CTNNBIP1* encodes a receptor for β-catenin, and their binding promotes the catabolism of β-catenin, which in turn puts the Wnt pathway in an inactive state (Fu et al., 2018). Its mutation has been shown to be associated with anterior segment dysgenesis 5 in the human disease database MalaCards (https://www.malacards.org/). *CLSTN1* is a member of the calsyntenin family, a subset of the cadherin superfamily. It can mediate the axonal anterograde transport of certain types of vesicles (Nagase et al., 1998)”.